# Respiratory Health Effects of Wildfire Smoke during Summer of 2018 in the Jämtland Härjedalen Region, Sweden

**DOI:** 10.3390/ijerph18136987

**Published:** 2021-06-29

**Authors:** Andreas Tornevi, Camilla Andersson, Ana Cristina Carvalho, Joakim Langner, Nikolai Stenfors, Bertil Forsberg

**Affiliations:** 1Section of Sustainable Health, Department of Public Health and Clinical Medicine, Umeå University, SE90187 Umeå, Sweden; bertil.forsberg@umu.se; 2Swedish Meteorological and Hydrological Institute (SMHI), Folkborgsvägen 17, SE60176 Norrköping, Sweden; camilla.andersson@smhi.se (C.A.); ana.carvalho@smhi.se (A.C.C.); Joakim.Langner@smhi.se (J.L.); 3Section of Medicine, Pulmonary Medicine, Department of Public Health and Clinical Medicine, Umeå University, SE90187 Umeå, Sweden; nikolai.stenfors@umu.se

**Keywords:** wildfires, forest smoke, PM_2.5_, chemistry transport model, health care visits, respiratory health, asthma

## Abstract

During the summer of 2018 Sweden experienced a high occurrence of wildfires, most intense in the low-densely populated Jämtland Härjedalen region. The aim of this study was to investigate any short-term respiratory health effects due to deteriorated air quality generated by the smoke from wildfires. For each municipality in the region Jämtland Härjedalen, daily population-weighted concentrations of fine particulate matter (PM_2.5_) were calculated through the application of the MATCH chemistry transport model. Modelled levels of PM_2.5_ were obtained for two summer periods (2017, 2018). Potential health effects of wildfire related levels of PM_2.5_ were examined by studying daily health care contacts concerning respiratory problems in each municipality in a quasi-Poisson regression model, adjusting for long-term trends, weekday patterns and weather conditions. In the municipality most exposed to wildfire smoke, having 9 days with daily maximum 1-h mean of PM_2.5_ > 20 μg/m^3^, smoke days resulted in a significant increase in daily asthma visits the same and two following days (relative risk (RR) = 2.64, 95% confidence interval (CI): 1.28–5.47). Meta-estimates for all eight municipalities revealed statistically significant increase in asthma visits (RR = 1.68, 95% CI: 1.09–2.57) and also when grouping all disorders of the lower airways (RR = 1.40, 95% CI: 1.01–1.92).

## 1. Introduction

Wildfires appears to be an increasing problem in different parts of the world, not least due to climate change [1,2,3,4,5]. Wildfires contribution to air pollution is significant, and has been estimated to generate 20% of the emissions of fine fraction particles (particulate matter with aerodynamic diameter ≤ 2.5 μm; PM_2.5_) in the USA and 30% in Canada [6]. The mixture of gaseous pollutants and levels of particulate matter (PM) in the ambient air are established risk factors for many health outcomes both in a short- and long-term perspective. Studies have linked deteriorated air quality to respiratory- and cardiovascular diseases, stroke, dementia and adverse pregnancy outcomes, and also associated with a shortened life expectancy [7,8,9,10,11,12]. Based on studies on the effect of PM_2.5_ on mortality rates it has been proposed that the global effect of emissions from wildfires are in the order of 340,000 deaths per year [13]. There is a growing body of health assessment studies and emissions from wildfires although the field is less studied than urban emissions which commonly originates from industries and combustion engines. Liu et al. identified 61 studies on the health effects of smoke from wildfires, which of 35 studies had measured levels on air quality where levels of particle matter were the most common way of indicating the level of exposure [14]. The content of PM_10_ (particulate matter ≤ 10 μm) in the studies was 1.2–10 times higher than normal during the episodes of wildfires.

The summer of 2018 Sweden experienced an unusual prevalence of wildfires, likely a consequence of an exceptionally prolonged period of dry weather and high temperatures [15]. Among the regions affected in 2018 were Gävleborg, Dalarna and Jämtland Härjedalen located in the northern half of Sweden. In general, large wildfires had not been common or widespread in Sweden, why the preparedness for the many large fires in 2018 was insufficient. Firefighters were tirelessly trying to contain and extinguish the fires, the armed forces assisted in some places, thousands of locals volunteered to help, and assistance in terms of helicopters and firefighting planes came from several countries including Norway and Italy [15].

The most intense wildfires occurred in the Jämtland Härjedalen region during the last weeks of July. Jämtland Härjedalen has a population of around 130,000 people and is one of the least densely populated regions in Sweden with 2.7 people per square kilometer. Jämtland Härjedalen includes eight municipalities with one larger city (Östersund) where around half of the population in the region is residing. The area of Jämtland Härjedalen is around 49,000 km^2^, which represents 12% of the total area in Sweden, and the majority of the area is forests.

The aim of this study was to investigate any presence of short-term respiratory health effects in the population exposed to decreased air quality caused by the wildfires in 2018, by studying fluctuations of daily number of health care contacts concerning respiratory illnesses. Exposure assessment studies on air pollution most commonly use air quality measurements from nearby monitoring stations. Suitable monitoring stations were however not available for this rather large region. Therefore, to identify periods with high exposure from wildfires across the region, simulated population-weighted levels of PM_2.5_ was generated by using a Multi-Scale Atmospheric Transport and Chemistry (MACTH) model.

## 2. Methods

### 2.1. Exposure Data Modelling

In the Jämtland Härjedalen region there was only one station with PM_2.5_ measurements; the European Monitoring and Evaluation Program (EMEP) Bredkälen station situated in the northern part of the region (Strömsund municipality, Figure 1B). This station was therefore not suitable for describing air quality for the entire Jämtland Härjedalen region.

The MATCH model [16,17,18] was applied to simulate exposure to wildfire smoke in the Jämtland Härjedalen region. Data were simulated for two periods; the summer of 2017 and 2018.

MATCH is a state-of-the-art, flexible, Eulerian, chemistry transport model. The model can be applied on a wide range of spatial scales and allows using different forcing meteorological input and emissions data. MATCH is one of the core models in operational air pollution forecasting for Europe at the Copernicus Atmosphere Monitoring Service (CAMS) [19]. The model is a building block in the MATCH Sweden system for environmental surveillance [20]. The model results are used to report to the European Union and for evaluation of air quality in Swedish municipalities. MATCH has a long history of research applications such as study of processes, future scenarios and historical mappings for understanding impacts on human health and ecosystems [21,22]. MATCH describes tropospheric ozone formation through photochemical reactions, atmospheric transformation and deposition of nitrogen and sulfur species and aerosol composition and dynamics. The photochemistry scheme in MATCH is based on the EMEP MSC-W chemistry scheme [23], with modified isoprene oxidation based on the so-called Carter-1 mechanism [24]. The aerosol scheme includes both primary particles and secondary particle formation and aerosol dynamics. The Secondary Organic Aerosols (SOA) description is based on the work by Bergström and Hodzic et al. [25,26]. The calculation of boundary layer height and the description of vertical transport in the boundary layer is based on surface heat and water vapour fluxes that are either diagnosed in the model or taken from the input meteorological data [16]. Dry deposition of gases and aerosols is modelled using a resistance approach, which depends on surface type and season; the deposition of gases to vegetated surfaces are coupled to soil moisture, temperature, vapour pressure deficit, and photo synthetically active radiation. The wet scavenging is assumed to be proportional to the precipitation intensity for most gaseous and aerosol components (following Simpson et al., 2012) [23].

The exposure modelling consisted in two nested MATCH simulations, with a “mother” domain covering a large part of Europe and an inner domain covering parts of Norway and Sweden including the Jämtland Härjedalen region (Figure 1A). The mother domain has a 0.1° × 0.1° grid cell horizontal resolution. The inner domain was defined to cover the region of interest and surrounding areas, with a grid resolution of 4 km × 4 km (Figure 1A).

The Global Fire Assimilation System (GFAS) daily product [27], available at 0.1° × 0.1° resolution (which approximately corresponds to 6 km × 12 km in central Europe), was used to account for atmospheric emissions from wildfires. A selection of compounds provided by GFAS were considered between June-August in 2017 and 2018; elemental carbon (EC), organic carbon (OC), total carbon (TC), nitrogen oxides (NOx), PM_10_, PM_2.5_, non-methane volatile organic compounds (NMVOC), carbon monoxide (CO), methane (CH_4_), and sulphur dioxide (SO_2_). These emissions were interpolated to the two different MATCH domains. The vertical distribution of the biomass burning emissions follows the work of Walter et al. [28]. The emissions were distributed using a parabolic function between the surface and the top of the plume, here fixed to 1500 m, based on the GFAS estimates for the summer. The daily emissions were distributed over the day using a Gaussian distribution with higher emissions during daytime and lower during night (following Kaiser et al. [29]). Two nested simulations were conducted for each year, one with GFAS emission included everywhere and one were the GFAS emissions were masked over a region covering almost all of Sweden. Anthropogenic emissions of NOx, SOx, NH_3_, NMVOC and CO were taken from the CAMS REG AP_v3.1/2016 dataset [30], at 0.1° × 0.5° (~6 km × 6 km in central Europe) for the European domain, while emissions from the Nordic WelfAir project emission inventory v5 on 1 × 1 km^2^ resolution, were used for the inner domain. 

Biogenic emissions of monoterpene, sesquiterpene and isoprene are calculated online in the model, following Simpson et al. [23], considering temperature at 2 m, radiation fluxes and vegetation cover.

The MATCH model was forced by the meteorological fields obtained from of the European Centre for Medium-Range Weather Forecasts (ECMWF) operational weather forecasting model (IFS), interpolated to the two domains. Land cover for the European domain were based on Corine land cover adjusted to the EMEP classes (as described by Simpson et al., 2012). For the Jämtland Härjedalen region, a newer Corine land cover data set (2012) was used to update the land use classes.

Hourly concentrations of PM_2.5_ was calculated from the modelled particle components, and the total PM_2.5_ and contribution due to wildfire emissions was separated using the runs with and without GFAS emissions in the inner domain. The geographically resolved PM_2.5_ concentrations were combined with gridded population data for the municipalities in the Jämtland Härjedalen region, collected from the National Population Register in order to calculate a population weighted exposure *E^k^*, according to $E^{k}=\overset{n^{k}}{\underset{i=1}{\sum }}P_{i}C_{i}/P_{k}$, where *i* denote a grid square among the total of *n* grid squares in municipality *k*, and *P* and *C* are population and PM_2.5_ counts, respectively. The modelled PM_2.5_ (daily maximum 1-h mean) data are illustrated in Figure 2.

### 2.2. Health Data

Information was obtained on the daily number of emergency room visits for respiratory diseases (diagnosis codes ICD-10: J00-J99) at Östersund Hospital, and acute visits to all primary health care centers and clinics located in the eight municipalities in the Jämtland Härjedalen region (Berg, Bräcke, Härjedalen, Krokom, Ragunda, Ströms, Åre and Östersund). Data covered to the period 2016 (1 January)–2018 (31 December) and was tabulated by date of contact, municipality of residence, health care unit, and diagnosis. Diagnoses were aggregated into all respiratory diseases (J00-J99), all upper and all lower respiratory diagnoses (J0, J3 and J1, J4, respectively), and asthma (ICD-10: J45, J46). Data was restricted to only include individuals with their home address registered in the region Jämtland Härjedalen, and aggregated according to municipality of residence.

A secondary source of health care contacts comprised daily number of phone calls to the National Healthcare Guide (commonly called: “1177”). Calls to the Healthcare Guide are classified by nurses according to predefined groups of problems, and for this study we obtained data classified as “breathing problems”. This data covered the period 1 January 2016–9 November 2018 and included only contacts from persons with their home address in any of the eight municipalities in the Jämtland Härjedalen region.

### 2.3. Epidemiological Analysis

A time series regression model was used to investigate whether the number of daily acute health care contacts (visits to primary health care centers and emergency rooms, or phone calls to the Healthcare Guide) was affected by decreasing air quality from the wildfires in 2018. The time series regression model adjusted for long-term trends in the outcome data by using a variable representing a cumulative count of observation days, and thereby the regression model aimed to investigate short-term variations. Long-term trends were adjusted with a smooth penalizing spline function (four degrees of freedom per year) and thereby able to adjust to varying seasonal pattern. Outcome data were assumed to be quassi-Poisson distributed and the regression models also adjusted for varying numbers of health care contacts depending on day-of-week, air temperature and relative humidity. For temperature and relative humidity, daily mean values were used and relationships were fitted with penalized spline functions (4 degrees of freedom) to allow for eventual non-linear associations.

The modeled particle data (daily maximum 1-h mean PM_2.5_) was used to determine exposure to smoke from wildfires. Since the simulated particle levels were generated only for two summer periods (June–September, 2017 and 2018), these data were used to define days with high levels of PM_2.5_ in the different municipalities. To define days with a clear contribution of smoke from wildfires in the ambient air, a cut off of at 20 μg/m^3^ was chosen (~98% percentile). This gave an indicator variable for ‘wildfire smoke days’ defined as of PM_2.5_ > 20 μg/m^3^, which were then compared with days with PM_2.5_ ≤ 20 μg/m^3^ in analysis. All days outside the simulated period of PM_2.5_ were assumed to be ≤ 20 μg/m^3^. This assumption was considered valid as data on PM_2.5_ from the EMEP station at Bredkälen indicated no other episodes during 2016–2018 with registered > 10 μg/m^3^ (24-h mean values). Additionally, the outcome variables (number of health care contacts) were studied in relation to the defined wildfire smoke days with a delay of up to 2 days. A lagged period of 0–2 days probably also captures any visits that were a result of wildfire smoke days that occurred during weekends, when visits to health care centers are often not possible.

The daily number of diagnoses and contacts were initially studied in relation to exposure separately for residents from each municipality. With the generated municipality-specific relative risks, a meta-estimate was calculated to generate a combined estimate for the total population in the entire region.

In an additional analysis, with aim to explore the concentration-response curve and any non-linear relationship between number of contacts for respiratory disease and levels of PM_2.5_. The analyses were performed on municipalities with a PM_2.5_ range large enough to make them meaningful, using a population-weighted maximum 3-day running mean population-weighted PM_2.5_ > 50 μg/m^3^ as inclusion criteria. A two-stage regression analyze was performed. In the first stage the outcome variables were adjusted for long-term trends, weekday patterns, temperature and relative humidity in an equally manner as described above (using all daily data during 2016–2018). In the second stage the adjusted data from the regression model (the residuals) was analyzed in relation to modeled PM_2.5_ (data covering summer periods of 2017 and 2018). Non-linearity between adjusted number of health care contacts and PM_2.5_ was examined by a penalized spline function with 5 degrees of freedom. An overall association between PM_2.5_ and healthcare contacts for the included municipalities was achieved by including a random intercept for *municipally* in the 1:st stage, along with municipality-specific adjustment for other covariates.

All analyzes were performed in the programming language R (version 4.03, The R Foundation for Statistical Computing, Vienna, Austria) with the packages *mgcv* (for penalizing splines) and *metaphor* (for meta-estimation of relative risks). For statistical tests, a significance level of 0.05 (two-sided) was used. Sensitivity analyzes were performed by varying the flexibility of the trend functions up to 12 degrees of freedom per year.

## 3. Results

During the period 2016–2018, inhabitants of the Jämtland Härjedalen region had 35,443 hospital emergency room visits and acute visits to primary health care centers for respiratory diseases (ICD-10: J00-J99). Of these visits, 92.3% where registered at health care centers. A total of 4613 phone calls due to ‘breathing problems’ were registered to Healthcare Guide. In Table 1 descriptive statistics about daily number of registrations on the studied groups of diagnoses are summarized for each municipally, where the City of Östersund had the highest number of diagnosis (*n* = 16,876, ICD-10: J00-J99) and Ragunda the lowest total numbers (*n* = 1185). Figure 3 illustrates all respiratory diagnoses per day from acute visits and shows a seasonal pattern with visits peaking during winter periods.

Defining the exposure of smoke from wildfires by days with daily maximum 1-h mean levels of PM_2.5_ > 20 μg/m^3^ resulted in a different number of days with wildfire smoke in the different municipalities. The municipality of Härjedalen was most exposed during nine days, and inhabitants in Strömsund and Ragunda was exposed only on one day. With the defined delay period of two consecutive days (lag 0–2), the number of health care contacts during 3–13 days (depending on the municipality) in the summer of 2018 was analyzed relative to other days during 2016–2018. Figure 4 illustrates the dates on which the population was exposed to PM_2.5_ > 20 μg/m^3^ (daily maximum 1-h mean) in the different municipalities.

The regression models estimated that days of wildfire resulted in a statistically significant increased number of diagnoses regarding asthma for the population in Härjedalen (Relative Risk (RR) = 2.6, 95% confidence interval (CI): 1.28–5.47). The population of Härjedalen was also exposed to most number of days (9 days) with 1-h maximum PM_2.5_ > 20 µg/m^3^ where on average the analyzed period of wildfire smoke reflected an increase in daily maximum 1-hr mean PM_2.5_ of 31 μg/m^3^. An increased number of diagnoses regarding ‘lower airways’ was also observed in the population of Strömsund (RR = 3.0, CI: 1.07–8.5) where the population where only exposed to one wild fire smoke day. Regarding other diagnostic groups, including calls to Healthcare Guide for ‘breathing problems’, no significant associations were observed in the different municipalities. For the population in Ragunda, no diagnoses were observed for *asthma* and *lower airways* during the defined wildfire smoke days (RR = 0, CI: NA). Similarly, for Berg municipally regarding diagnoses of *upper airways*, and these (invalid) estimates were excluded when calculating RR for the whole region (meta-analysis). All estimates for each municipally are shown in Table 2.

The meta-analyses for all eight municipalities estimated a significantly increased risk for health care visits regarding diagnoses about asthma (RR = 1.68, 95% CI: 1.09–2.57) and for diagnoses on lower airways as a group (RR = 1.4, 95% CI: 1.01–1.92). For the whole region wildfire smoke days on average responded to an increase a daily maximum 1-hr mean PM_2.5_ of 28 μg/m^3^.

In the analyze aiming to investigate non-linearity (or potential effects above certain PM_2.5_ levels) between PM_2.5_ and health care contacts, no such pattern were observed or could be determined. Significant linear increases in number of health care contacts with increase in PM_2.5_ were however observed among the population in Härjedalen municipality regarding asthma and diagnoses on lower airways as a group. Diagnoses about lower airways were also observed to increase linearly with increasing levels of PM_2.5_ for the population living in Östersund. In Figure 5 results from the two stage analysis is illustrated. In an overall analyze on number of health care visits in the most exposed municipalities in terms of modelled levels of PM_2.5_, a significant association was observed diagnoses concerning lower airways (Figure 6).

## 4. Discussion

Wildfires occurred during the summer of 2018 in the Jämtland Härjedalen region of Sweden. The concurrent decrease in ambient air quality was associated with a significant increase in the number of regional health care contacts for asthma. A significant increase was also observed when combining all respiratory diagnoses regarding lower airways including asthma. We focused on potential impacts on acute respiratory problems in order to get the best statistical power to determine associations and as we had a small number, relative to studies from other countries, of days with smoke from wildfire and exposed individuals.

In 2015 Liu et al. published an overview of peer-reviewed scientific studies from 1986 and later regarding impacts of wildfire smoke on health in exposed communities. They identified 61 epidemiological studies linking wildfire and human health in scales ranged from one city population of about 55,000 to the entire globe. Respiratory disease was the most frequently studied health condition, and in over 90% of 45 such studies there was a significant increase in respiratory morbidity associated with the smoke exposure [14].

Another review included a total of 53 studies and concluded the epidemiological studies to have too much differences regarding the exposure data to enable a meta–analysis [4]. However, the effects on asthma and the respiratory system were considered well–established, while the results on cardiovascular and mental effects were considered varied and uncertain. In a follow-up analysis, adding articles published up to August 2018, the authors conclude that there is clear evidence that smoke from fires impairs the health of people with asthma, but that the results for chronic obstructive pulmonary disease (COPD) are less consistent [31].

A more recent systematic review with the aim to estimate the association between short-term exposure of fine particulate matter from wildfires and asthma-related outcomes [32] included 20 studies for a quantitative meta-analysis. PM_2.5_ was positively associated with a 6% increase of asthma hospitalizations (RR = 1.06, 95% CI: 1.02–1.09) and a 7% increase in emergency department visits for asthma (RR = 1.07, 95% CI: 1.04–1.09) per 10 μg/m^3^ increase in the daily mean value (or the chosen time window). For hospitalizations, the largest effect size was found for lag 0 (same day) compared to other single–day lags, (RR = 1.07, 95% CI: 1.01–1.13), and there seemed to a be statistically significant effect both in one day (RR = 1.03, 95% CI: 1.01–1.05), and two days after exposure (RR = 1.03, 95% CI: 1.01–1.05). For emergency room visits, the effect was not statistically significant at lag 1 and 2, however positive (RR = 1.02).

In our data on asthma visits, a wildfire smoke day (PM_2.5_ (daily maximum 1-h mean) exposure >20 μg/m^3^) resulted in an observed RR = 1.68 as a mean for the same day as exposure and two following days. With the mean increase in exposure of 28 μg/m^3^ as maximum 1-h mean such a day, it seems that applying the reported meta-estimates for lag 0, 1 and 2 for emergency room visits for asthma would have resulted in less than 68% increase in patients, even if we assume that also lag days often had elevated exposure levels.

One strength and novel approach in our study is to use geocoded population data and model population-weighted exposure at municipal level to study the daily number of cases for each municipality.

A common limitation in epidemiological studies is the lack of information on potential confounding factors. However, with time-series studies risk factors that do not change over short periods of time (e.g., smoking, occupation, disease predisposition or pet exposure) cannot co-vary with fluctuations in daily exposure and cannot result in confounding. Air pollutants not originating from the wildfires could be seen as potential confounders, but in this sparsely populated region with usually very clean air it is very unlikely. Information of the age of the cases in our study population could have provided information of susceptible age-groups. Surprisingly, a recent publication on the health effects of the Australian bushfire season 2019/2020 found that age > 65 years was associated with a significantly lower risk of adverse health effects. The authors presumed that older people are more cautious, more indoors, and thus less exposed than younger people [33].

This study has shown that even relatively limited wildfires with short periods of increased air pollution in a sparsely populated region may increase the risk of respiratory morbidity in the affected population. Simple measures to mitigate the effect of wildfires would be public health messaging to increase the adherence to respiratory medications, such as inhalers, and attentiveness within the health care against an increased burden of patients in need of care.

## 5. Conclusions

A chemistry transport model was used to simulate levels of PM_2.5_ in a sparsely populated region in Sweden (Jämtland Härjedalen) aiming to quantify the degree of air pollution caused by wildfires that occurred during the summer of 2018. For the population living in the region, population-weighted PM_2.5_ levels were associated with an increase in asthma diagnoses and when analyzing all diseases in the lower airways as a group. With climate change and more periods of high temperatures, wildfires can be an increasing problem in many parts of the world, and this study shows that relatively limited wildfires with short periods of decreased air quality can increase risk of respiratory morbidity and an increased burden on number of patients at hospitals and health care centers.

## Figures and Tables

**Figure 1 ijerph-18-06987-f001:**
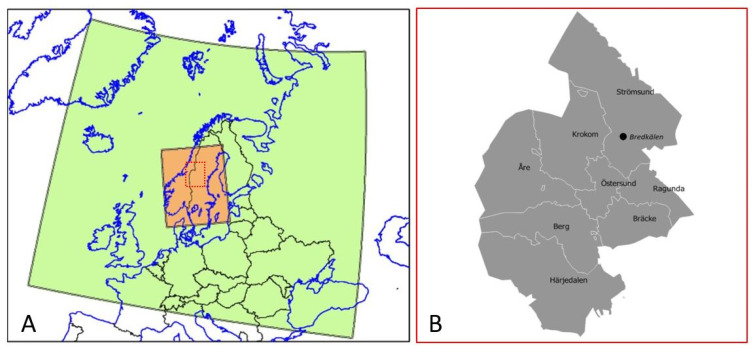
MATCH domains covering northern Europe (panel **A**, green), at a 0.1° × 0.1° resolution, and the inner domain covering Jämtland Härjedalen and surrounding areas (panel **A**, orange), at 4 km × 4 km resolution. In panel (**B**) the Jämtland Härjedalen region municipalities are illustrated together with an approximate location of the EMEP station Bredkälen.

**Figure 2 ijerph-18-06987-f002:**
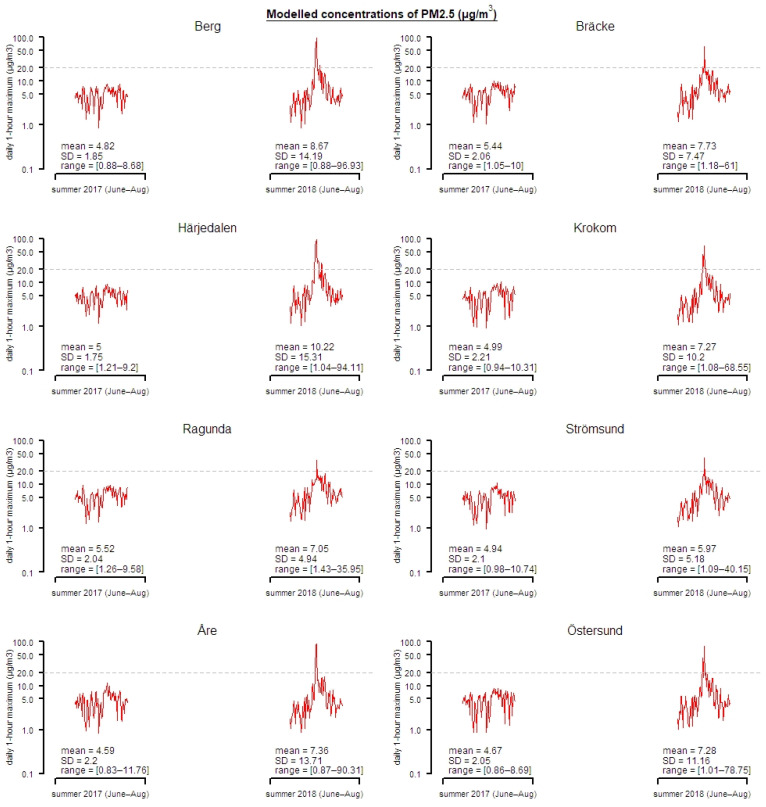
Modelled population-weighted concentrations of PM_2.5_ (daily maximum 1-h mean values) for the municipalities in region Jämtland Härjedalen June to August 2017 and 2018. *Y*-axis are on a log scale, and for each municipality, the total mean value (mean), standard deviation (SD) and minimum and maximum values (range) are printed.

**Figure 3 ijerph-18-06987-f003:**
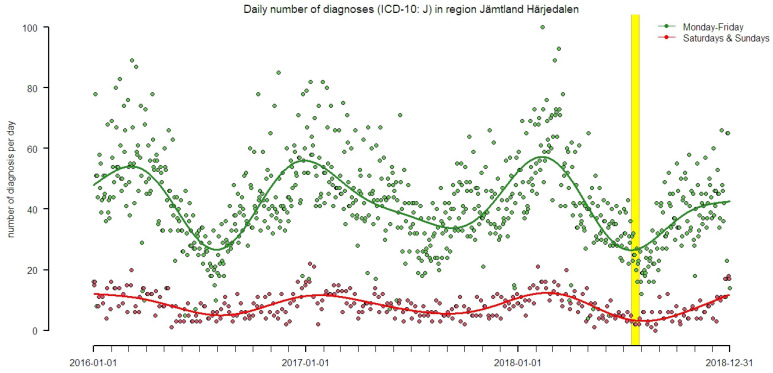
Daily number of acute visits for the diagnoses ICD-10: J for the population in Jämtland Härjedalen registered at healthcare centers in the region’s 8 municipalities, and at Östersund hospital. Data is separated by weekdays (green) and weekends (red) to illustrate the difference in frequency, and the yellow shading represents the time period were most of the wildfires occurred (15–28 of July 2018). Smooth spline functions with 4 degrees of freedom per year illustrates long-term trends (green and red lines).

**Figure 4 ijerph-18-06987-f004:**
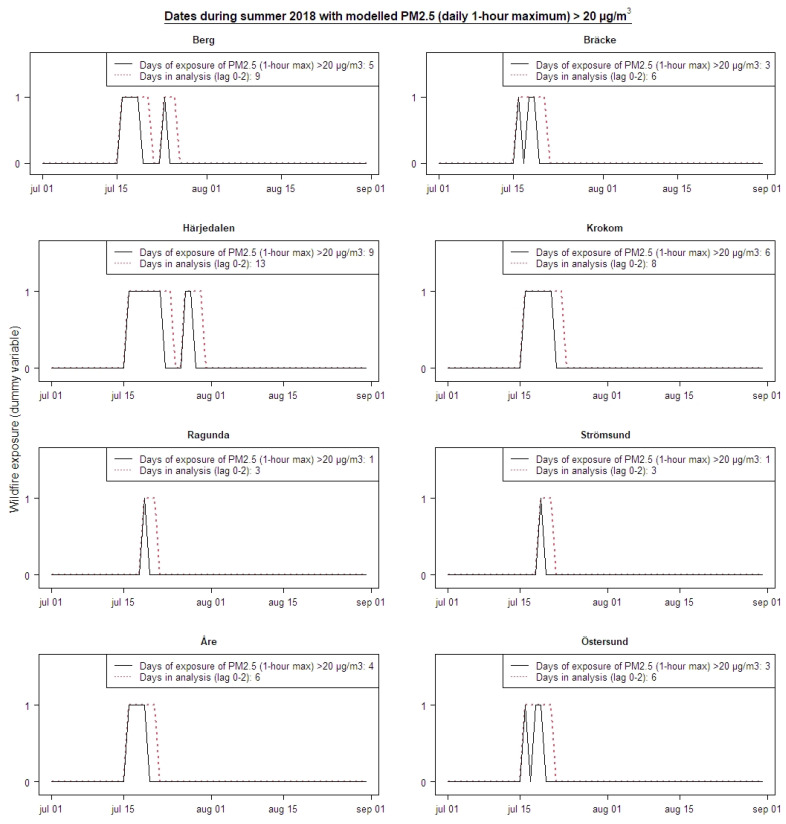
Dates of exposure to wildfire smoke defined as maximum 1-h mean value of PM_2.5_ > 20 µg/m^3^ for 8 municipalities in the summer of 2018 (black), and dates when the number of healthcare contacts was analyzed in regression models according to a delay period of 2 days (lag 0–2) (red, dashed).

**Figure 5 ijerph-18-06987-f005:**
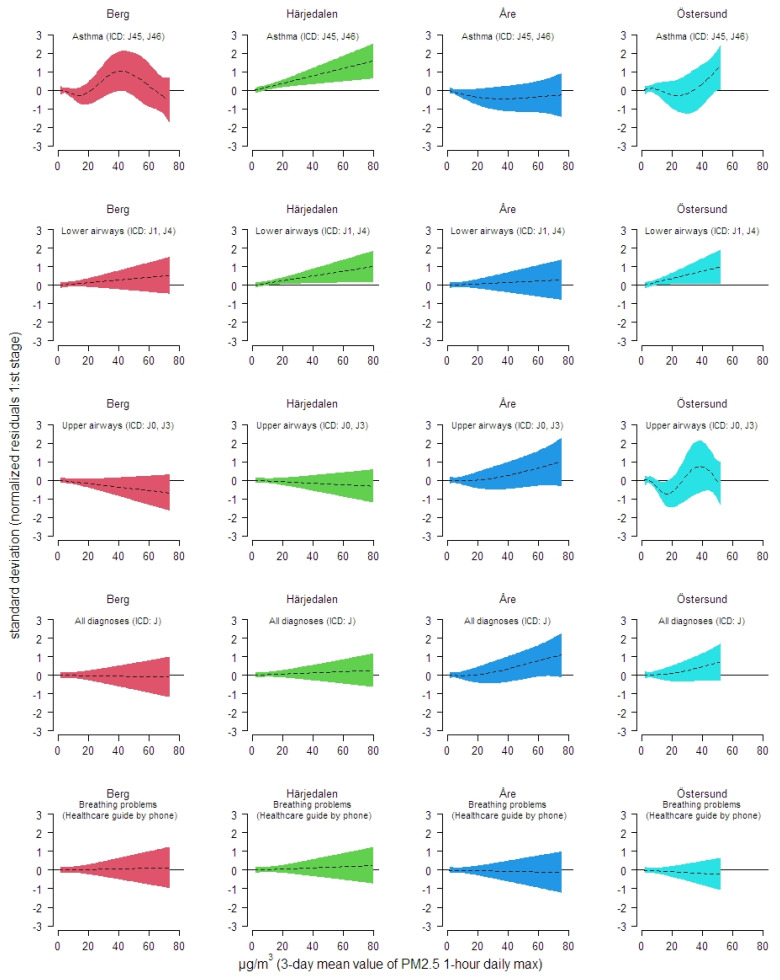
Associations between modelled exposure to PM_2.5_ and different healthcare contacts (trend- and covariate adjusted residuals) in the four municipalities in the Jämtland Härjedalen region which were most exposed of smoke from wildfires summer of 2018.

**Figure 6 ijerph-18-06987-f006:**
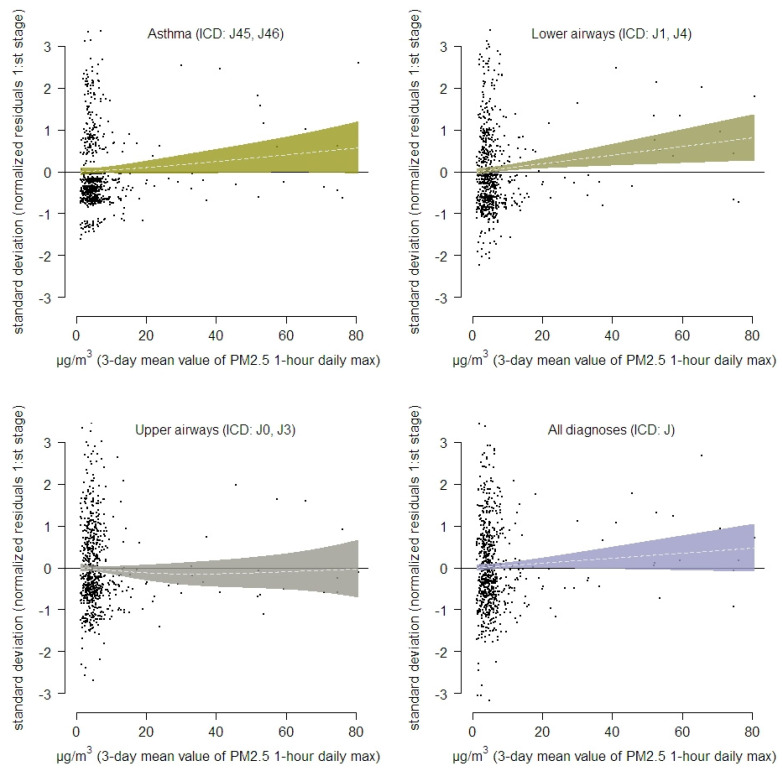
Overall associations between modelled exposure to PM_2.5_ and different healthcare visits including the municipalities of Berg, Härjedalen, Åre and the city of Östersund.

**Table 1 ijerph-18-06987-t001:** Descriptive statistics regarding studied groups of diagnoses in the eight municipalities and their population count (pop) as of 31 December 2018 (Statistics Sweden, SCB); total number of diagnoses during the period 1 January 2016–31 December 2018 (Healthcare Guide by phone until 2018-11-09). N = Total number of diagnoses/phone calls, {daily average} [max–min].

Municipality/(Population Count)	Asthma (ICD: J45, J46)	Lower Airways (ICD: J1, J4)	Upper Airways (ICD: J0, J3)	All Respiratory (ICD: J00-J99)	Breathing Problems (Healthcare Guide)
Berg (pop = 7097)	N = 145	N = 412	N = 665	N = 1364	N = 191
	{0.13}/[0–2]	{0.38}/[0–3]	{0.61}/[0–6]	{1.24}/[0–6]	{0.18}/[0–3]
Bräcke (pop = 6376)	N = 162	N = 522	N = 889	N = 1657	N = 220
	{0.15}/[0–2]	{0.48}/[0–5]	{0.81}/[0–6]	{1.51}/[0–8]	{0.21}/[0–3]
Härjedalen (pop = 10147)	N = 383	N = 985	N = 2176	N = 3757	N = 315
	{0.35}/[0–5]	{0.9}/[0–6]	{1.99}/[0–11]	{3.43}/[0–14]	{0.3}/[0–3]
Krokom (pop = 14858)	N = 412	N = 1002	N = 2235	N = 3627	N = 461
	{0.38}/[0–3]	{0.91} /[0–6]	{2.04}/[0–9]	{3.31}/[0–16]	{0.44}/[0–4]
Ragunda (pop = 5343)	N = 121	N = 351	N = 637	N = 1185	N = 242
	{0.11}/[0–3]	{0.32}/[0–4]	{0.58}/[0–5]	{1.08}/[0–9]	{0.23}/[0–6]
Strömsund (pop = 13253)	N = 361	N = 929	N = 1739	N = 3217	N = 435
	{0.33}/[0–3]	{0.85}/[0–6]	{1.59}/[0–12]	{2.94}/[0–17]	{0.42}/[0–4]
Åre (pop = 11529)	N = 475	N = 859	N = 2421	N = 3760	N = 244
	{0.43}/[0–4]	{0.78}/[0–7]	{2.21}/[0–10]	{3.43}/[0–15]	{0.23}/[0–4]
Östersund (pop = 63227)	N = 1459	N = 3864	N = 10981	N = 16876	N = 2505
	{1.33}/[0–8]	{3.53}/[0–16]	{10.02}/[0–32]	{15.4}/[0–51]	{2.4}/[0–12]

**Table 2 ijerph-18-06987-t002:** Estimated relative risks (RR) and 95% confidence intervals [95% CI] associated with exposure to smoke from wildfires in the individual municipalities and combined into a meta-estimate.

Municipality	Asthma (ICD: J45, J46)	Lower Airways (ICD: J1, J4)	Upper Airways (ICD: J0, J3)	All Respiratory (ICD: J00-J99)	Breathing Problems (Healthcare Guide by Phone)
	RR [95% CI]	RR [95% CI]	RR [95% CI]	RR [95% CI]	RR [95% CI]
Berg	0.95 [0.11–7.96]	1.34 [0.53–3.42]	0 [0–Inf]	0.62 [0.26–1.50]	1.49 [0.19–11.88]
Bräcke	1.92 [0.46–8.04]	1.05 [0.32–3.41]	0.88 [0.27–2.86]	0.91 [0.41–2.06]	1.18 [0.15–9.32]
Härjedalen	2.64 [1.28–5.47]	1.48 [0.74–2.99]	0.69 [0.34–1.36]	1.01 [0.64–1.58]	1.28 [0.44–3.70]
Krokom	0.39 [0.05–2.85]	0.22 [0.03–1.64]	0.62 [0.29–1.34]	0.50 [0.25–1.00]	0.84 [0.19–3.76]
Ragunda	0 [0–Inf]	0 [0–Inf]	0.90 [0.12–6.81]	0.49 [0.06–3.80]	1.02 [0.12–8.47]
Strömsund	2.59 [0.61–11.07]	3.02 [1.07–8.54]	0.96 [0.28–3.28]	1.53 [0.71–3.34]	2.70 [0.57–12.72]
Åre	0.37 [0.05–2.91]	1.15 [0.39–3.35]	1.75 [0.9–3.38]	1.56 [0.89–2.73]	3.31 [0.73–15.04]
Östersund	1.44 [0.70–2.97]	1.41 [0.84–2.36]	0.86 [0.59–1.25]	1.12 [0.82–1.52]	0.82 [0.39–1.72]
Meta–estimate	1.68 [1.09–2.57]	1.40 [1.01–1.92]	0.90 [0.70–1.69]	1.04 [0.85–1.27]	1.20 [0.75–1.90]

## Data Availability

The data presented in this study are available on request from the corresponding author.

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
