# Peer review of "Respiratory Health Effects of Wildfire Smoke during Summer of 2018 in the Jämtland Härjedalen Region, Sweden"

_ijerph, 2021, doi:10.3390/ijerph18136987_

Round 1
Reviewer 1 Report
The paper entitled Respiratory health effects of wildfire smoke during summer of 2018 in region Jämtland Härjedalen, Sweden have been structured well enough and the results are interesting. the author needs to expand the introduction section.
Author Response
Thank you for your suggestion!
We have extended the introduction with more information about the wildfires and information about the region Jämtland Härjedalen.
Reviewer 2 Report
Reviewer
Abstract
Lines - 28-29-30-31- Modelled levels of PM2.5 were obtained for the two summer periods of 2017 and 2018. Potential health effects of wildfire related levels of PM2.5 were examined by studying daily health care contacts concerning respiratory problems in each municipality in a quasi-Poisson regression model, adjusting for long-term trends, weekday patterns and weather conditions.
Comment:
How many people were evaluated in the period from 2017 to 2018?
And the age and sex of the people evaluated? What about work activity?
Introduction
Lines 44-45 - Studies have linked deteriorated air quality to respiratory - and cardiovascular diseases (REF),
Comment:
Please, what does REF mean?
Methods
Exposure data modelling
Comment:
Lines 75-76- …the EMEP station Bredkälen situated in the northern part of the region (Strömsund municipally).
Please, what does EMEP mean?
Comment:
Line 84- …air pollution forecasting for Europe in CAMS
Please, what does CAMS mean?
Comment:
Line 86- …..to report to the EU and for evaluation
Please, what does EU mean?
Lines 93-94- The aerosol scheme includes both primary particles and secondary particle formation and aerosol dynamics (Andersson et a., 2015).
Comment:
Lines 93-94- (Andersson et a., 2015) - et al.
- Andersson, C., et al., MATCH-SALSA–Multi-scale Atmospheric Transport and CHemistry model coupled 405 to the SALSA aerosol microphysics model–Part 1: Model description and evaluation. Geoscientific Model 406 Development, 2015. 8(2): p. 171-189.
What is the reference number? 18?
Lines 120-121-122- The daily total wildfire emissions for a selection of compounds (EC, OC, PM10, 120 PM2.5, TC, NOX, NMVOC, CO, CH4, and SO2), during June- August in 2017 and 2018. 121 Anthropogenic emissions of NOx, SOx, NH3, NMVOC and CO were taken from….
Comment:
EC, OC, PM10, 120 PM2.5, TC, NOX, NMVOC, CO, CH4, and SO2…
SOx,.
Please, describe these abbreviations in full.
Lines 145-146 - Figure 1. Modelled population weighted concentrations of PM2.5 (daily mean values) for the municipalities in region Jämtland Härjedalen June to August 2017 and 2018.
Comment:
How many people were evaluated in the period from 2016 to 2018?
Health data
Line 153 - Data covered to the period 2016 (1 January) - 2018 (31 December) and………..
Comment:
How many people were evaluated in the period from 2016 to 2018?
And the age and sex of the people evaluated? What about work activity?
Epidemiological analysis
Line 190 - …during 206-2018 with registered…..
Comment:
Please correct for - 2016 - 2018
Ethical Consideration
I couldn't read at work.
Results
Lines 223-224- During the period 2016-2018, inhabitants of region Jämtland Härjedalen had 35443 hospital emergency room visits and acute visits to primary health care centers for respiratory diseases (ICD-10: J00-J99).
Comment:
Why the bold?
Discussion
Lines 342-343 - Information of the age of the cases in our study population could have provided information of susceptible age-groups.
Comment:
Is this one of the limitations of this study?
Conclusions
Lines 355-356-357-358- A chemistry transport model was used to simulate levels of PM2.5 from the wildfires. The population weighted PM2.5 levels in region Jämtland Härjedalen in 2018 were associated with an increase in asthma visits as well as visits for all diseases in the lower airways among the population living in the region.
Comment:
Lines 357-358- ….as well as visits for all diseases in the lower airways among the population living in the region
We can conclude in this paper that ....... as well as visits for all diseases in the lower airways among the population living in the region.
What are ALL the diseases?
References
Comment:
Are the references only with the first author and et al.?
They are correct?
References are missing the DOI
Thank you
Author Response
Reviewer
Thank you for your review!
Abstract
Lines - 28-29-30-31- Modelled levels of PM2.5 were obtained for the two summer periods of 2017 and 2018. Potential health effects of wildfire related levels of PM2.5 were examined by studying daily health care contacts concerning respiratory problems in each municipality in a quasi-Poisson regression model, adjusting for long-term trends, weekday patterns and weather conditions.
Comment:
How many people were evaluated in the period from 2017 to 2018?
And the age and sex of the people evaluated? What about work activity?
All residents living in region the Jämtland Härjedalen are eligible to be included. However, we only have the number of patients per day, not their personal identity number. We did not use any excluding factors (like age or gender). All visits to primary health centers/hospital with diagnosis codes ICD-10 J (and subgroups, for example asthma ICD-10: J45,46) were included in the analyses. The same individual could also have been included more than once over the time period. We did not use age specific numbers because of the low total frequency. We have update table 1 with number of residents for each municipally (as of 2018-12-31).
We don’t have any information about work activity, but we adjust in the models for weekend and work days.
Introduction
Lines 44-45 - Studies have linked deteriorated air quality to respiratory - and cardiovascular diseases (REF),
Comment:
Please, what does REF mean?
Sorry, this was a mistake and ‘(REF)’ has now been excluded.
Methods
Exposure data modelling
Comment:
Lines 75-76- …the EMEP station Bredkälen situated in the northern part of the region (Strömsund municipally).
Please, what does EMEP mean?
All abbreviations have now been spelled out in the updated manuscript.
Comment:
Line 84- …air pollution forecasting for Europe in CAMS
Please, what does CAMS mean?
All abbreviations have now been spelled out in the updated manuscript.
Comment:
Line 86- …..to report to the EU and for evaluation
Please, what does EU mean?
All abbreviations have now been spelled out in the updated manuscript.
Lines 93-94- The aerosol scheme includes both primary particles and secondary particle formation and aerosol dynamics (Andersson et a., 2015).
Lines 93-94- Corrected to (Andersson et al., 2015)
Comment:
- Andersson, C., et al., MATCH-SALSA–Multi-scale Atmospheric Transport and CHemistry model coupled 405 to the SALSA aerosol microphysics model–Part 1: Model description and evaluation. Geoscientific Model 406 Development, 2015. 8(2): p. 171-189.
What is the reference number? 18?
Yes, it was 18, thank you for noticing this. The reference has been listed earlier in the text, and this notation one is now erased.
Lines 120-121-122- The daily total wildfire emissions for a selection of compounds (EC, OC, PM10, 120 PM2.5, TC, NOX, NMVOC, CO, CH4, and SO2), during June- August in 2017 and 2018. 121 Anthropogenic emissions of NOx, SOx, NH3, NMVOC and CO were taken from….
Comment:
EC, OC, PM10, 120 PM2.5, TC, NOX, NMVOC, CO, CH4, and SO2…
SOx,.
Please, describe these abbreviations in full.
All abbreviations have now been spelled out in the updated manuscript.
Lines 145-146 - Figure 1. Modelled population weighted concentrations of PM2.5 (daily mean values) for the municipalities in region Jämtland Härjedalen June to August 2017 and 2018.
Comment:
How many people were evaluated in the period from 2016 to 2018?
Please see answer above (first question)!
Health data
Line 153 - Data covered to the period 2016 (1 January) - 2018 (31 December) and………..
Comment:
How many people were evaluated in the period from 2016 to 2018?
And the age and sex of the people evaluated? What about work activity?
Please see answer above!
Epidemiological analysis
Line 190 - …during 206-2018 with registered…..
Comment:
Please correct for - 2016 - 2018
Yes, now corrected, Thank you!
Ethical Consideration
I couldn't read at work.
Results
Lines 223-224- During the period 2016-2018, inhabitants of region Jämtland Härjedalen had 35443 hospital emergency room visits and acute visits to primary health care centers for respiratory diseases (ICD-10: J00-J99).
Comment:
Why the bold?
A mistake, this was not supposed to be in bold.
Discussion
Lines 342-343 - Information of the age of the cases in our study population could have provided information of susceptible age-groups.
Comment:
Is this one of the limitations of this study?
A limitation is surely the small population in the study, which, for example, limits analyses on subgroups, for example the elderly.
Conclusions
Lines 355-356-357-358- A chemistry transport model was used to simulate levels of PM2.5 from the wildfires. The population weighted PM2.5 levels in region Jämtland Härjedalen in 2018 were associated with an increase in asthma visits as well as visits for all diseases in the lower airways among the population living in the region.
Comment:
Lines 357-358- ….as well as visits for all diseases in the lower airways among the population living in the region
We can conclude in this paper that ....... as well as visits for all diseases in the lower airways among the population living in the region.
What are ALL the diseases?
With ‘all’ diseases in the lower airways we mean all diagnostic codes which belong to ICD-10 J1 and J4. We have rewritten and extended the Conclusion part. We hope this is more clear now.
References
Comment:
Are the references only with the first author and et al.?
They are correct?
References are missing the DOI
We hope we got the reference style is correct now, Thank you!
Thank you
Reviewer 3 Report
Please follow the comments in the attached document to improve the manuscript. There are some methodological issues that need to be clarified. Conclusions should also be improved.

Author Response
Thank you for your review!
Please see attachment

Reviewer 4 Report
As a reviewer I have the following remarks.
- The paper is well written and presented.
- Abstract. Line 27. Please say “concentrations of fine particulate matter (PM2.5)”…
- Abstract. Lines34-36 . It’s good to define RR and CI in the first use, One solution is as follows: relative risk, RR =2.64, 95% confidence interval: [1.28-5.47], and later use (RR = 1.68, [1.09-2.57]), This convention is used in Table 2 [ ] denotes 95% CI.
- Line 43: “levels of particulate matter (PM)” – as PM is not defined. Also PM2.5 ls less or equal (line 42).
- Line 45: “and cardiovascular diseases (REF),- ? It will change references numeration.
- Line 54: From the Internet: PM10 : inhalable particles, with diameters that are generally 10 micrometers and smaller; - thus also =.
- Line 79: “Data was simulated”. Data is plural (datum).
- Line 108. 0.1º x 0.1º - it will be good to add (~10 km x 10 km) or something similar as later is used 4x4km.
- Line 156: “and asthma (ICD-10: J45, J46): - say, for asthma only these codes were used?
- Line 178: “For temperature and relative humidity” – also daily means as for PM2.5?
- Line 216: “the programming language R” – it will be good at least specify version + company name for R, say TR (ver. 4.06,..)
- Fig 2. Please specify what kind of lines are fitted (red and green).
- Line 271: “(RR = 1.68, CI: [1.09-2.57])” – a new notation CI:[ ]?
Thank you
Author Response
As a reviewer I have the following remarks.
1. The paper is well written and presented.
Thank you for your review, much appreciated!
2. Abstract. Line 27. Please say “concentrations of fine particulate matter (PM2.5)”…
Yes, we followed your advice.
3. Abstract. Lines34-36 . It’s good to define RR and CI in the first use, One solution is as follows: relative risk, RR =2.64, 95% confidence interval: [1.28-5.47], and later use (RR = 1.68, [1.09-2.57]), This convention is used in Table 2 [ ] denotes 95% CI.
Thanks, we have followed your advice.
4. Line 43: “levels of particulate matter (PM)” – as PM is not defined. Also PM2.5 ls less or equal (line 42).
We have clarified this, and changed the to “less or equal”
5. Line 45: “and cardiovascular diseases (REF),- ? It will change references numeration.
Sorry, this ‘(REF)’ was a mistake. Shouldn’t have been left there.
6. Line 54: From the Internet: PM10 : inhalable particles, with diameters that are generally 10 micrometers and smaller; - thus also =.
We change it to be less or equal.
7. Line 79: “Data was simulated”. Data is plural (datum).
Correct! Now it’s written in plural (‘were’)
8. Line 108. 0.1º x 0.1º - it will be good to add (~10 km x 10 km) or something similar as later is used 4x4km.
Yes, this is a bit complicated as it depends how far from the equator the grid area is.
We have added that “( which approximately corresponds to ~ 6 km x 12 km in central Europe)”
9. Line 156: “and asthma (ICD-10: J45, J46): - say, for asthma only these codes were used?
Yes, these were the only ICD codes used to define asthma diagnoses in our data.
10. Line 178: “For temperature and relative humidity” – also daily means as for PM2.5?
Yes, we used daily means for temp and RH, and this is now clarified in the text (new line number 203)
11. Line 216: “the programming language R” – it will be good at least specify version + company name for R, say TR (ver. 4.06,..)
We now spelled out version and name (The R Foundation for Statistical Computing, version 4.03)
12. Fig 2. Please specify what kind of lines are fitted (red and green).
Agree, now we specified this in the figure text. “Smooth spline functions with 4 degrees of freedom per year illustrates long-term trends (green and red lines)” The figure is also modified a bit.
13. Line 271: “(RR = 1.68, CI: [1.09-2.57])” – a new notation CI:[ ]?
Sorry, this was also a typo, thank you for noticing. We also notice some inconsistency with CI notations. Now corrected.
Round 2
Reviewer 1 Report
The author adds few lines in the Introduction section but did not incorporate citations in the highlighted lines. Before going to publish the paper, authors need to provide appropriate citations.
Author Response
Thank you again for your review. We have corrected/adjusted and included the reference to the highlighted lines in the Introduction.
Reviewer 3 Report
The changes have improved the manuscript
Author Response
Thank you again for your review. We agree that the changes improved the manuscript.